# Innovative Techniques of Processing Human Milk to Preserve Key Components

**DOI:** 10.3390/nu11051169

**Published:** 2019-05-24

**Authors:** Aleksandra Wesolowska, Elena Sinkiewicz-Darol, Olga Barbarska, Urszula Bernatowicz-Lojko, Maria Katarzyna Borszewska-Kornacka, Johannes B. van Goudoever

**Affiliations:** 1Laboratory of Human Milk and Lactation Research at Regional Human Milk Bank in Holy Hospital, Medical University of Warsaw, Department of Neonatology, 63A Zwirki i Wigury St., 02-091 Warsaw, Poland; olga.barbarska@wum.edu.pl; 2Human Milk Bank, Ludwik Rydygier’ Provincial Polyclinical Hospital in Torun, Torun, 53-59 St. Jozef St., 87-100 Torun, Poland; elena.darol@wszz.torun.pl (E.S.-D.); ursber@interia.pl (U.B.-L.); 3Department of Neonatology and Intensive Care Unit, 1st Faculty of Medicine, Medical University of Warsaw; Karowa 2, 00-315 Warsaw, Poland; maria.borszewskakornacka@gmail.com; 4Department of Pediatrics, Emma Children’s Hospital, Amsterdam UMC, Vrije Universiteit Amsterdam, P.O. Box 7057, 1007 MB Amsterdam, The Netherlands; h.vangoudoever@amsterdamumc.nl

**Keywords:** breastfeeding, high pressure processing, newborn, holder pasteurization, human milk, donor milk

## Abstract

Human milk not only contains all nutritional elements that an infant requires, but is also the source of components whose regulatory role was confirmed by demonstrating health-related deficiencies in formula-fed children. A human milk diet is especially important for premature babies in the neonatal intensive care unit (NICU). In cases where breastfeeding is not possible and the mother’s own milk is insufficient in volume, the most preferred food is pasteurized donor milk. The number of human milk banks has increased recently but their technical infrastructure is continuously developing. Heat treatment at a low temperature and long time, also known as holder pasteurization (62.5 °C, 30 min), is the most widespread method of human milk processing, whose effects on the quality of donor milk is well documented. Holder pasteurization destroys vegetative forms of bacteria and most viruses including human immunodeficiency virus (HIV) herpes and cytomegalovirus (CMV). The macronutrients remain relatively intact but various beneficial components are destroyed completely or compromised. Enzymes and immune cells are the most heat sensitive elements. The bactericidal capacity of heat-pasteurized milk is lower than that of untreated milk. The aim of the study was for a comprehensive comparison of currently tested methods of improving the preservation stage. Innovative techniques of milk processing should minimize the risk of milk-borne infections and preserve the bioactivity of this complex biological fluid better than the holder method. In the present paper, the most promising thermal pasteurization condition (72 °C–75 °C,) and a few non-thermal processes were discussed (high pressure processing, microwave irradiation). This narrative review presents an overview of methods of human milk preservation that have been explored to improve the safety and quality of donor milk.

## 1. Introduction

Recent findings have confirmed the therapeutic properties of human milk components and have left no doubt that it constitutes an indispensable part of newborns’ nutritional treatment, especially premature babies with very low (VLBW) and extremely low (ELBW) birth weight. Initiating lactation in preterms’ mothers and maintaining mother’s milk supply for NICU infants remains challenging. Donor milk has become the standard way of feeding newborns who cannot receive the milk of their biological mothers [1]. Donor milk administration to prematurely born children needing longer hospital stays requires necessary collecting procedures, freezing, storage and pasteurization. This may considerably lower its nutritional and therapeutic values. So far, the best characterized and commonly used method of human milk preservation is holder pasteurization. However, the knowledge about the significant negative impacts of this method on the active components in human milk results in the development of novel techniques that could better preserve nutritional and non-nutritional factors.

The aim of this narrative review was to analyze the best characterized and current methods of human milk processing in the context of minimizing qualitative and quantitative losses of its bioactive elements and ensuring safety in clinical situations.

## 2. Materials and Methods

The literature review included electronic searches of MEDLINE (January 2000–January 2019), EMBASE (January 2000–January 2019) and conference proceedings. The electronic search used the following text words and MeSH terms: donor milk, human milk, breast milk, banked milk, (human milk OR donor milk) AND pasteurization [–] (human milk OR breast milk) AND, preservation (human milk OR breast milk) AND holder pasteurization (human milk OR breast milk) AND, high-pressure (human milk OR breast milk), UV–treatment (human milk OR breast milk), microwave (human milk OR breast milk). Reference lists of the previous reviews and relevant studies were examined. Research that had been reported only as abstracts were eligible for inclusion if sufficient information was available from the report.

## 3. Results and Discussion

Microbiological safety is the primary criterion that has to be fulfilled as far as food for newborns is concerned. Therefore, firstly we focus on the effectiveness of the proposed human milk processing methods in human milk pathogen elimination (Table 1).

Microbiota associated with human milk mostly reflects the mother’s health status but contributes to both maternal and infant homeostasis [2]. Small amounts of bacteria from the nipple and the areola, such as *Escherichia coli, Serratia marcescens,* and *Pseudomonas aeruginosa* are often present in human milk. Moreover, due to milk flow back from the newborn’s mouth cavity into the mammary ducts, *Streptococcus sp.* and *Staphylococcus sp.* are present in mother’s milk in physiological conditions [3]. On the other hand, bacteria connected with mastitis, namely *Staphylococcus aureus, Streptococcus agalacitiae* and *Corynebacterium,* are potentially dangerous for newborns. Human milk may also become contaminated with other bacteria such as *Listeria monocytogenes, Enterobacter cloacae* and *Klebsiella pneumoniae* as a result of improper expressing, collection and storage [4]. Antibodies against numerous pathogenic bacteria, fungi and viruses are present in human milk and passed to the breastfed infant to prevent mother-to-child disease transmission.

Due to this fact, even though the milk of an infected mother does contain pathogens, bioactive components of the natural food effectively prevent the entry of bacteria and viruses into the digestive tract of a child. Free fatty acids and monoglycerides (products of triglycerides lipolysis) found in human milk exhibit anti-fungal activity [22]. The bacteriostatic properties of human milk depend on its specific antibodies but also result from toxin neutralization by cellular components of the immune system, as well as bacterial translocation blocking by intestinal mucosa. Therefore, the binding of *Streptococcus pneumoniae* and *Escherichia coli* to their proper receptors is inhibited by human milk oligosaccharides (HMO) and in the case of *Campylobacter jejuni*, by fucosyl oligosaccharides. Kappa–casein is a specific ligand for *Helicobacter pylori*. Neutralization of *Escherichia coli, Clostridium difficile*, *Salmonella enterica* and *Shigella* enterotoxins is the reason why breastfeeding successfully protects against bacterial diarrhea [23]. Also, the binding of rotaviruses by stable sedimentation of 46 kDa glycoprotein (lactadherin), resulting in the alleviation of infectious symptoms, has been confirmed [24]. Therefore, with proper hygiene during the process of expressing and storage conditions, bacterial infections of infants caused by the milk of their mothers are extremely rare. Donor milk that is intended for a child other than a biological child undergoes more restrictive handling [25]. Human milk may also be a source of cytomegalovirus (CMV), hepatitis B virus (HBV) and human T lymphotropic retrovirus (HTLV I and II), as well as HIV 1, 2 infection. Even though transmission through human milk is rare, the pasteurization parameters of milk intended for children should ensure the deactivation of pathogenic viruses HIV, HTLV, CMV, rubella virus and herpes virus (HCV) (hepatitis virus is relatively heat-resistant) [26].

### 3.1. Methods of Human Milk Processing

Low temperature long time (LTLT) pasteurization, also known as the holder method (HoP), is considered to be the standard for human milk pasteurization. Milk is incubated for 30 min at 62.5 °C in a water bath or other device that ensures effective heating [27,28]. Modern milk pasteurizers guarantee precise measurement of the temperature inside the bottle, automatic control of the process and the possibility of efficient and safe milk cooling to low temperature (usually 4 °C) [29]. Other methods of high temperature pasteurization at 72–75 °C, the so-called HTST (high temperature short time (HTST) or FHP flash-heat pasteurization (FHP), are used to effectively eliminate microorganisms [30,31]. Flash pasteurization applies especially in countries with a high risk of HIV infection. It was adopted as a simple, universally accessible method and does not require any sophisticated equipment. Today a simple, portable, low resource piece of equipment for FHP is available on the market. Escuder-Vieco and coworkers [11] recently demonstrated an HTST equipment for the continuous processing of human milk that could be adopted in milk banks. International research has shown that the high temperature short time method was effective in eradicating HIV, *Escherichia coli*, *Staphylococcus aureus* and *Streptococcus A* and *B*, while protecting the nutritional composition of human milk including important vitamins [32,33,34]. Also, investigations of methods combining thermal processing with homogenization techniques, for example ultrasound, are being carried out at present. Other physical factors such as UV, microwaves, electric pulses, and high pressures as alternatives to the traditional thermal process of inactivating microorganisms in human milk have been tested [35,36,37,38,39,40,41].

One of the most promising non-thermal method is high pressure processing (HPP), which is becoming a more and more popular method of food preservation on an industrial scale. It seems to be more efficient in achieving microbial purity of human milk, while still preserving its maximal therapeutic value [7,42,43].

### 3.2. Temperature and High Pressure Influence on Microbial Purity of Human Milk

The main goal of human milk pasteurization is to remove pathogens that might possibly be the cause of an infection when donor milk is given to an infant as an alternative to formula feeding (in accordance with WHO, UNICEF and AAP guidelines) [1,44]. The efficiency of pathogen elimination such as *Mycobacterium tuberculosis* by heating milk to 62.5 °C was first checked in cow’s milk [28]. Later research confirmed the possibility of removing vegetative forms of numerous types of bacteria present in human milk as well as the risk that endosporic forms and toxins will survive, like in the case of *Bacillus cereus,* still remain [45]. This is why in many European countries, the microbial purity of human milk is controlled twice, before and after pasteurization, eliminating milk with a level of *Staphyloccocus aureus* above 10^4^ CFU (colony forming units) before pasteurization and/or an overall bacteria count above 10^5^ CFU. In the absence of a unified acceptance criteria for milk before pasteurization with respect to microbial purity, in order for milk to be given to someone other than the biological child of the donor, vegetative forms of bacteria must be eliminated (Table 1).

In the case of Ebola virus, Marburg and Zika viruses, inactivation happens only by using thermal processing (HoP). So far, there is no data on whether HTST or HPP are equally effective as HoP in eliminating these viruses [15,16].

HTST pasteurization (72 °C, 5–16 s) seems to be more effective than the holder method in eliminating bacteria and viruses with lipid envelopes (HIV, HTLV), as well as model viruses for HCV and hepatitis B virus that cannot be otherwise deactivated [20,46]. Even high temperatures do not completely eliminate viruses without lipid envelopes like Parvovirus B19, but apart from experimental conditions, no risk of infection transferred through human milk has been reported [31]. HTST was reported to efficiently destroy CMV infectivity in a process at 72 °C for 5 s [17]. Microwave radiation at high-power settings has proved to inactivate CMV in human milk as well [19] (Table 1).

Recently, Escuder-Vieco reported that HTST processing at 72 °C for at least 10 s efficiently destroyed all vegetative forms of microorganisms present in raw milk [11]. However, sporulated *Bacillus sp*. survived this thermal process. In earlier studies Klotz and coworkers did not find a difference in the reduction of naturally present microorganisms in raw milk after HTST in comparison to HoP treatment [10]. High pressure processing (HPP) is becoming an increasingly popular method for food preservation on an industrial scale. The selective effects of high pressures allows the conditions (pressure, temperature, presence of water) to be chosen, in order to selectively destroy pathogenic cells and preserve more of the valuable human milk components. HPP is a successful method of eliminating Gram-positive and Gram-negative bacteria, though vegetative cells are more effectively destroyed than endosporic forms. The destruction of *Listeria monocytogenes, Eschericha coli, Staphylococcus aureus, Staphylococcus agalactiae* and *Salmonella spp*. within the pressure range of 300–400 MPa has been proved safe for a lot of milk proteins with active hormones and enzymes. Viruses such as HIV and CMV are also deactivated [7,31,43]. HPP was shown to be even more effective in comparison to holder pasteurization in the elimination of inoculated microbiological flora (total viable count microorganisms, *S. aureus*) of raw milk [12]. A recent study showed that high pressure processing can be an effective method of eliminating bacteria which produce spores, like *Bacillus cereus* [13]. Foods pasteurized by means of the HPP method are freer of microbes than products of thermal processing. Therefore, HPP may possibly be used in the future in milk banks, providing nutrition for newborns with special dietary needs.

### 3.3. The Influence of Processing on Bioactive Components of Human Milk

The effects of different types of processing (HoP, HTST, HPP) on human milk bioactive factors are shown in Table 2. Methodology discrepancies between included research papers, especially in biological factors given as an indicator of milk bioactivity, did not allow for a systematic review of the topic. This is a main limitation of our study.

The bacteriostatic effect of human milk, resulting from the presence of lysozyme, lactoferrin and lactoperoxidase, is an extremely valuable property. The first experiments conducted to preserve human milk by Wills revealed a reduction in lactoferrin and lysozyme activity after holder pasteurization to 27% and 67%, respectively [28]. Other investigations demonstrated lactoferrin activity was completely destroyed and a changing effect on the lysozyme, depending on the experimental conditions [6]. The latest research showed a reduced content and activity of lysozyme, lactoperoxidase and lactoferrin, while a shorter incubation time in higher temperature (75 °C, 15 s versus 62.5 °C, 30 min) proved to have better results as far as antibacterial properties were concerned [30]. Moreover, recent studies have shown a 40% reduction of lactoferrin concentration in HoP-treated milk [12]. This could explain the several-fold increase of bacteria in milk after pasteurization in comparison to fresh or frozen milk [49]. Taking both methods of thermal processing into consideration, longer incubation at lower temperatures (62.5 °C, 30 min.) appears to be more suitable. It is due to this fact that, in comparison to HTST which reduces the bacteriostatic effect of milk to 36%, holder pasteurization preserves 52% of bactericidal properties (as controls non-pasteurized milk caused a 70% reduction in *E. coli* growth). Subsequent storage at 4 °C for 48 and 72 h did not affect the bactericidal effect, which had been confirmed by earlier investigations [5].

Tests of the effect on *Listeria innocua* antigens, measured with ELISA, confirmed the superiority of holder pasteurization. The bacteriostatic effect was 1.8 units in non-pasteurized milk, 1.3 activity units (28% decrease in activity) in milk pasteurized by LTLT, and 1.1 units (39% decrease) by HTLT [48]. It is difficult to compare the effect of HTST on the concentration and activity of lysozymes and enzymes with bacterial activity. Peila and coworkers pointed out that the results are divergent due to different methods that were used for HTST in each study [79]. Measurements of antioxidant levels in pasteurized milk showed reduced antioxidant activity in human milk in high temperatures. It is important to mention that shorter incubation at higher temperatures (75 °C 15 s) caused a smaller loss of human milk components that neutralize free radicals [54].

Bertino demonstrated no changes in oligosaccharide composition after holder pasteurization, therefore donor milk may be a source of these valuable biomolecules [56]. Temperatures within the range of 62.5–63 °C do not influence the presence of gangliosides, the receptors for *Bifidobacteria* adhesion proteins [62]. What is more, the process does not change the amount of biogenic polyamines, like spermine and spermidine, and their regulatory functions [58].

Thermal processing depletes human milk of hormones such as adiponectin, insulin and leptin. In a study by Ley et al. the reduction of adiponectin and insulin was 33% and 46%, respectively [60]. Another study showed a reduction of HMV adiponectin, insulin and leptin concentration of 38%, 32% and 88% respectively, and also incubation at 57 °C reduced the number of bioactive peptides like leptin [12,61]. In contrast to temperature treatment, in four tested variants of high pressure: 600 MPa, 100 + 600 MPa, 200 + 400 MPa and 200 + 600 MPa the reduction in HPP content was 64%, 61%, 2% and 57%, respectively. Adipokines, key metabolism regulation components (leptin, insulin, HMV adiponectin) of infant growth and body composition, have also been targeted in studies. Compared to the thermal method, HPP lead to better retention of leptin and insulin. In the case of leptin, high pressure processing even caused an increase in concentration (148–186%) [12].

Holder pasteurization seems to have no significant influence on the overall content of fats in milk, leaving the activity of unsaturated LC-PUFA fatty acids like DHA and AA unaltered [64,65]. High pressures also do not deactivate free fatty acids, only at high pressure combined with high temperature (600–900 MPa 50 °C–80 °C) were changes in the proportion of some fatty acids observed [62,63,77,78]. However, the latest research indicated a 5.5% decrease in the overall content of lipids before and after pasteurization, and the content of the analyzed components differed significantly between the samples [80]. It is important to have in mind that the absence of lipase, which hydrolyzes fatty acids in non-pasteurized milk releasing them from triacylglycerols, may also influence the obtained values of fatty acid levels in pasteurized milk. That explains, reported in some studies, the slightly elevated level of these fats in pasteurized milk [65]. The analysis of lactose isomerization markers and Maillard reaction markers showed that lactulose and furosine concentrations were lower in HTST-treated milk than after holder pasteurization. In this study, *γ*-glutamil transpeptidase was also shown to be thermoresistant, but both methods completely destroyed alkaline phosphatase [11]. In most studies, ALP and lipases activity are completely reduced by both thermal methods [10]. So far, only one study showed higher concentrations of bile salt-stimulated lipase (BSSL) and total lipase activity in milk treated with HTST in comparison with HoP [50]. High hydrostatic pressure is also effective in preserving BSSL, the retention of lipase in milk samples after using the HPP method was 80–85% [13].

Earlier research demonstrated that holder pasteurization at 62.5 °C does not influence the content of vitamins A, D, E, B2 in human milk and vitamin B12 is reduced to 48% in comparison to non-pasteurized milk [67,69]. Folic acid, vitamins B1, B2, B6 and C seem to be resistant to the influence of even higher temperatures (up to 72 °C) [27]. Later research demonstrates vitamin A content decreases in milk after holder pasteurization and, consequently, its deficiency in pasteurized milk (36.6 +/– 26.1 mg/100 mL) [68]. Holder pasteurization reduces the amount of vitamin C (36%), folic acid (31%) and vitamin B6 (15%) in human milk [27,63]. The higher temperature (75 °C) used for HIV deactivation in human milk also lowers the concentration of vitamin B6 (92%) and B2 (59%), without influencing vitamin A. At the same time, high temperature processing increases the effect of vitamin C, B12 and folic acid [66]. Regardless of the processing method, active alpha-, gamma- and delta- tocopherols are preserved in human milk, in contrast to vitamin C which, as the latest studies proved, is reduced by at least 16% in high temperatures. Compared to thermal processing, HPP had no significant effect on the content of vitamin C and E [63,77]. A study by Vieira and coworkers indicated that human milk processed within the temperature range of 62–62.5°C has a significant impact on the overall content of proteins (a median decrease of protein content in pasteurized versus non-pasteurized milk of 4%) [80]. HTST pasteurization appears to have less influence on the activity of individual proteins in human milk, which was confirmed by proteomic studies [50]. Among various classes of antibodies present in human milk, IgA are relatively stable, whereas IgG are deprived of a large part of their activity and IgM are completely deactivated after both HoP and HTST pasteurization [47,69]. At first, Goldbloom et al. showed that HTST had no significant effect on IgA [47], but a later study presented that 5 s at 72 °C resulted in loss of IgA similar to this after HoP [27]. The use of HPP resulted in a 36% decrease in IgA content in milk samples [13]. Recent studies have shown that HoP had a statistically significant effect on IgG (49% reduction) and that retention of this immunoglobulin after HPP was in the range of 30% to 82%, depending on the high pressure applied [12]. Bile salt-stimulated lipase (BSSL) and other enzymes such as lactoperoxidase and amylase are also deactivated by temperature [49,50,65]. Immunological cells present in milk are destroyed and the amount of other components influencing host defense mechanisms, for example the soluble form of the CD14 antigen, is lowered, with an unaffected number of mannose-binding lectins [71,80].

Numerous growth factors present in fresh human milk remain active also after holder pasteurization, for example EGF (epidermal growth factor) or TGFβ and TGFα (transforming growth factor) [18,72]. Unfortunately, holder pasteurization significantly deactivates proteins such as IL-10, erythropoietin, IFN-γ, TNF-α, IL-1, IL-6, IL-13, HGF [73]. Wesolowska has recently reported that compared to holder pasteurization (89% reduction), HPP led to a better retention of HGF [12]. Cytokines from the group of insulin-like growth factors, IGF-1 and IGF-2, and their binding proteins, IGF-BP 2 and 3, are destroyed in the process of long incubation at 62°C, while short pasteurization at high temperatures does not affect them (72°C, 5 s) [18].

High pressure processing appears to be the most effective in preserving protein activity. The protective effect of high pressures depends on the presence of hydrogen bonds and secondary structure of proteins (structure of beta-sheet is more pressure-resistant than alpha helix). Protein denaturation below 400 MPa is a reversible process due to the effect of high pressure on the weakest hydrogen bonds and Van der Waals force, while preserving the integrity of covalent bonds and the structure ensuring enzymatic activity. The selective effect of high pressures allows the conditions (pressure, temperature, presence of water) to be chosen in order to selectively destroy pathogenic cells and preserve more of the valuable components of human milk. It was confirmed by results regarding the stability of a large group of relatively heat-resistant cytokines (EGF, TGF-*β*2, TNF-RI, TGF-*β*1 and IL-8) at 400 MPa, as well as statistically significant in the activity of temperature-sensitive cytokines in milk processed at high pressures, (IL-6, IL-13, TNF-*α* and IL-10, HGF), and high pressure processing of human milk also allows, to a greater extent, the activity of IgA antibodies to be preserved. Within the range of 400 to 600 MPa and at a temperature of 12 °C, the effect of IgA alters slightly (activity decreases to 88% for 500 MPa and 69% for 600 MPa). At 400 MPa, the activity of IgA in human milk remains intact in comparison to a 72% decrease in traditionally pasteurized milk (62.5 °C, 30 min). The use of 350 MPa pressure at 38 °C in certain conditions led to a 36% reduction in the content of IgA [13]. Also, the activity of lysozyme is unaffected by high pressures or thermal processing [8,13,43].

## 4. Conclusions

Mother’s own fresh expressed milk is an extremely valuable source of components with both nutrient and bioactive activities, such as bacteriostatic factors, oligosaccharides, vitamins and growth factors. The combined effect of the nutritional and bioactive components decides on the short-term and long-term health benefits of a human milk-based diet in the preterm newborn population. Some of the latest research has shown a partial activity of human milk after holder pasteurization for important biological implications such as anti-infective properties, immune components, microbiota and growth factors [79,81,82].

Improvement of the techniques allowing the preservation of the bioactivity of donor milk is profitable in the context of the health effects related to child feeding choice. Proven and safe techniques adopted from the food industry have created opportunities to minimize losses as a consequence of human milk processing. The new methods applied to human milk should be at least as effective as the old ones to ensure microbiological safety.

Data evaluating the effectiveness of interventions with thermal pasteurized donor milk are not clear [83,84,85,86,87], and there are no clinical trials concerning new techniques. Therefore, studies with human participants are needed to be carried out with the most promising new techniques of human milk processing in comparison to holder pasteurization.

It is essential to bear in mind if there is a real possibility of introducing new equipment into the market and its usage in routine milk bank services [88].

## Figures and Tables

**Table 1 nutrients-11-01169-t001:** Studies of Holder Pasteurization, High-Temperature Short-Time, High Pressure Processing and Microwave Irradiation HoP, HTST, HPP, MI on microbiological and viral components of human milk.

Tested Component	HoP	HTST	HPP	MI	References
Bacteriostatic effect on *E.coli* and *L. innocua*	48% reduction 28% reduction	64% reduction 39% reduction	not studied	not studied	[5,6]
Inactivation of selected microorganisms *(L. monocytogenes S. agalactiae, E. coli, S. aureus*)	inactivation	not studied	inactivation	not studied	[7]
Inactivation of selected microorganisms (*Enterobacteriaceae*)	inactivation	not studied	inactivation	not studied	[8]
Inactivation of selected microorganisms (*S. aureus ATCC 6538, Enterobacteriaceae*)	not studied	not studied	inactivation	not studied	[9]
Antibacterial efficacy (Coagulase-negative staphylococci, Gram- negative bacteria, Enterococcus species)	reduced bacterial counts	reduced bacterial counts	not studied	not studied	[10]
Microbiological quality (vegetative forms of microorganisms present in raw milk samples)	destroyed commensal and contaminant vegetative microorganisms except *Bacillus sp*.	destroyed all vegetative forms of microorganisms except *Bacillus sp*. and *E. faecalis*	not studied	not studied	[11]
Inactivation of selected microorganisms (*S.aureus*)	inactivation	not studied	inactivation	not studied	[12]
Inactivation of selected microorganisms (*S.aureus, B. cerues*)	partial inactivation	not studied	inactivation	not studied	[13]
Inactivation of selected microorganisms (*E.coli, P. aeruginosa, S. aureus, S. epidermidis*)	inactivation	not studied	not studied	inactivation	[14]
Ebola Virus	inactivation	not studied	not studied	not studied	[15]
Marburg Virus	inactivation	not studied	not studied	not studied	[15]
Zika virus	inactivation	not studied	not studied	not studied	[16]
CMV	Inactivation destroy viral infectivity	destroy viral infectivity	not studied	inactivation	[17,18,19]
HTLV HIV	inactivation	inactivation	not studied	not studied	[20]
HPV high-risk (types 16 and 18), low-risk (type 6)	inactivation	not studied	not studied	not studied	[21]

HoP–Holder Pasteurization, HTST–High-Temperature Short-Time, HPP–High Pressure Processing, MI–Microwave Irradiation CMV–cytomegalovirus, HTLV–human T lymphotropic virus, HIV–human immunodeficiency virus.

**Table 2 nutrients-11-01169-t002:** Effects of different type of processing (Holder Pasteurization, High-Temperature Short-Time, High Pressure Processing and Microwave Irradiation) on human milk factors.

Factor	HoP	HTST	HPP 400–600 MPa 5–30 min, 12 °C 37 °C	MI	References
	Activity Loss/Reduce of Concentration	Activity Loss/Reduce of Concentration	Activity Loss/Reduce of Concentration	Activity Loss/Reduce of CONCENTRATION	
Lactoferrin	63–100% 44–80% 64–83% 80% 60% 39% 61% 80% (65 °C)	11% 68% 0–14% 71%	* decrease (600 MPa, 15 min) 21–44% (200 + 400–600 MPa, 10 min) 300 MPa–650 MPa 15 min, 20 °C 300, 400, 500, 600 MPa, 15 min, 20 °C 9–48% 3–7%	ns	[5,6,10,12,13,27,30,31,33,47,48,49,50,51,52]
Lysozyme	* decrease 21–67% (reduction concentration) 33–76% 35% * increase stable 48%	55% 28% * increase	107% 400 MPa 30 min 14–25% 0–5% <5%	* decrease	[5,6,7,10,13,28,30,33,39,53]
Antioxidant activity (glutathione, glutathione peroxidase, malonedialdehyde, superoxide dismutase, TAC)	* decrease in glutathione concentration (about 50%), GPx activity (near 67%) and TAC (about 58%) reduced GPx	Decrease	no significant changes in TAC, reduction of AsA >11% (200 MPa, −20°C)	SOD and GPx activity temporary increased during microwave heating	[41,53,54,55]
Oligosaccharides	stable	Stable	ns	ns	[33,56,57]
Biogenic amines	stable	Ns	ns	ns	[58];
Glucocorticoids (cortisol, cortisone)	not significantly affected				[59]
Adiponectin HMV Adiponectin	34% 33%	Ns	62–98%	ns	[12,60]
Insulin	46% 33%	Ns	5–18%	ns	[12,60]
Leptin	40% (57 °C, 30 min) 78%	Ns	48–90% (* increase in leptin concentration)	ns	[12,61]
Medium-chain saturated fatty acids	no change in content	Ns	no change in content	ns	[62,63]
Long-chain unsaturated fatty acids	no change in content, slight decrease of oleic acid content	Ns	no change in content	**ns**	[62,63]
Polyunsaturated *fatty acids n* = 3, *n* = 6 Linolenic acid	no change in content	Ns	no change in content	ns	[62,63,64,65]
Folic acid	36%	* increase	ns	ns	[66];
Vitamin A	Stable decrease from 55.5 mg/100 mL to 36.6 mg/100 mL, about 34%	Stable	ns	ns	[66,67,68]
Vitamin B1(thiamine)	ns	Stable	ns	ns	[27]
Vitamin B2	stable	stable 41% 59%	ns	ns	[27,66,67]
Vitamin B6	15%	stable 18% 59%	ns	ns	[27,66]
Vitamin B12	48%	increase	ns	ns	[66,69]
Vitamin C	36% 16% 20–36% 35%	stable * increase	no change <5% (200 MPa, −20 °C)	ns	[27,41,63,66]
Vitamin D	stable	Ns	ns	ns	[67]
Vitamin E (tocopherol)	stable	Ns	ns	ns	[63,67]
Lysine	higher content of available lysine	Stable	ns	ns	[50]
IgM	complete deactivation decrease in content	Ns	ns	ns	[47,70]
IgA	20–100% 27% 21.1% decrease in content 57% (65 °C) 98% 49% 56%	20% 5% 74,8% 57%	0% (400 mPa) 13% (500 MPa) 32% (600 mPa) 5 min 12°C 40% 400, 500, 600 MPa 5 min, 12 °C 0–31% 17%	no significant effect	[8,10,13,33,39,43,50,51,52,70]
IgG	34–100% decrease in content 49%	33%	18–70%	ns	[12,47,70]
Alkaline phosphatase	complete loss	94%	ns	ns	[10]
Lipoprotein Lipase	complete loss 99%	stable 99%	15–20% stable		[10,13,50,65]
Lactoperoxidase	50–88%	stable	ns	ns	[49]
Amylase	15%	ns	ns	ns	[65]
Mannose-binding lectin	stable	ns	ns	ns	[71]
CD 14 (soluble)	88%	ns	ns	ns	[71]
TGF β1TGF β1	decrease <1%	ns	stable	ns	[72,73,74]
TGF α	decrease <6% stable	ns	ns	ns	[70,72,73]
IL-10	* decrease in content	substantial decrease	decrease (400 MPa, 5 min, 12 °C) * decrease (500 MPa) no presence at 600 MPa	ns	[33,62,73,74]
Erythropoietin	decrease	ns	ns	ns	[73]
IFN-	decrease in content	ns	ns	ns	[62]
TNF-α	* decrease in content	ns	decrease (400 MPa), * decrease (500 MPa, 600 MPa)	ns	[62,74]
TNF-RI	increase in content	ns	increase (400 MPa), increase (500 MPa), * increase (600 MPa)	ns	[74]
IL-1 α	* decrease in content substantial decrease	ns	ns	ns	[62]
IL-2	decrease in content	ns	ns	ns	[62]
IL-4	decrease in content	ns	ns	ns	[62]
IL-5	decrease in content	ns	ns	ns	[62]
IL-12p70	decrease in content	ns	ns	ns	[62]
IL-13	*decrease in content	ns	increase (400 MPa), * decrease (500 MPa) * decline (600 MPa)	ns	[62,74]
IL-8	25% increased content * increase of activity * increase	not studied * increase	increase (400 MPa), * increase (500 MPa) * increase (600 MPa)	ns	[33,62,74]
IL-6	* decrease of activity	ns	increase (400 MPa, 500 MPa, 600 MPa)	ns	[74]
HGF	33% 89%	ns	3–66%	ns	[12,62]
EGF	stable	stable	stable	ns	[18]
IGF-1	39%,	stable	ns	ns	[18]
IGF-2	9.9%	stable	ns	ns	[18]
IGF-BP2	19.1%	stable	ns	ns	[18]
IGF-BP3	7%	stable	ns	ns	[18]
Free nucleotide monophosphates (AMP, GMP, CMP, TMP)	stable	ns	400, 500, 600 MPa 5 min (without temp. control) stable or increased content	ns	[75]
Volatile profile	modified the volatile profile (lipid oxidation, Maillard reaction)	ns	HPP at 400 or 600 MPa for 3 min preserved the original volatile compounds of human milk	ns	[76]
**High Pressure Thermal Processing**					
Fatty acids, cytokines, leukocytes and immunoglobulins (IgM, IgA and IgG)	ns	ns	300–900 MPa, temp. 50–80 °C, for 1 min minimal effect on the levels of IL-12, IL-17 and IFN-γ. loss of leukocytes cells, only the treatments at 300 MPa and 50 °C maintained certain levels of Igs (IgM 25% loss, IgA 52% and IgG 0%)	ns	[77]
**Factor**	**HoP**	**HTST**	**HPP** **400–600 MPa** **5–30 min, 12 °C 37 °C,**	**MI**	**References**
	**activity loss/reduce of concentration**	**activity loss/reduce of concentration**	**activity loss/reduce of concentration**	**activity loss/reduce of concentration**	
Tocopherols, fatty acids, cytokines (IL-6, IL-8, IL-10, IL-12 (p70), IL-17, IFN-γ, TNF-α and MCAF/MCP-1)	* decreased the levels of α-, γ-and δ-tocopherol	ns	600 MPa * decreased the levels of α-, γ- and δ-tocopherol, reduction proportions of some key fatty acids, not affect on the levels of IL-6, IL-8 and TNF-α.	ns	[78]
IgA, IgM, IgG,	IgA 20% IgM 51% IgG 23%	ns	no/small effect on Igs, stable	ns	[42]
volatile profile	ns	ns	300–900 MPa, 50–80 °C * modified volatile profile	ns	[76]

HoP–Holder Pasteurization, HTST–High-Temperature Short-Time, HPP–High Pressure Processing, MI–Microwave Irradiation, MPa–Megapascal, GPx–glutathione peroxidase, TAC–total antioxidant capacity, AsA–ascorbic acid, SOD–superoxide dismutase, HMV–high molecular weight, IgA–immunoglobulin A, IgM–immunoglobulin M, IgG–immunoglobulin G, CD 14–cluster of differentiation 14, TGF *β_1_*–transforming growth factor beta 1, TGF *β_2_*–transforming growth factor beta 2, TGF *α*–transforming growth factor alpha, IL10–interleukin 10, IFN-*γ*–interferon gamma, TNF *α*–tumor necrosis factor alpha, TNF-RI–tumor necrosis factor receptor, IL-1*α*–interleukin 1 alpha, IL-2–interleukin 2, IL-4–interleukin 4, IL-5–interleukin 5, IL-12p70–interleukin 12, IL-13–interleukin 13, IL-8–interleukin 8, IL-6–interleukin 6, IL 17–interleukin 17, HGF–hepatocyte growth factor, EGF–epidermal Growth Factor, IGF1–insulin-like growth factor, *IGF2*–insulin-like growth factor 2, IGF-BP2–insulin like growth factor binding protein 2, IGF-BP3–insulin like growth factor binding protein 3, AMP–adenosine monophosphate, GMP–guanosine monophosphate, CMP–cytidine monophosphate, TMP–thymidine monophosphate, MCAF/MCP-1–monocyte chemotactic and activating factor/monocyte chemoattractant protein-1,* statistically significant (*p* < 0,05), ns–not studied.

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
