# Peer review of "Innovative Techniques of Processing Human Milk to Preserve Key Components"

_nutrients, 2019, doi:10.3390/nu11051169_

Reviewer 1 Report

This is an interesting and timely paper that describes the effect of various methods to sterilize mother's milk for use as donor milk. The authors did a thorough review of the literature and summarized the findings beautifully. The tables are helpful and really show how the various methods affect human milk composition and nutrient/bioactivity. The synthesis of the literature into this paper really provides a wonderful addition to what is known and compels the reader to not accept the status quo of holder pasteurization as the last word, that we need new methods already available through the food industry to improve the preservation of the nutritional and bioactive components of breast milk to ensure the best donor milk possible for our most at-risk infants.

My only suggestions are grammatical and formatting issues, which I have summarized below:

Line 26: own mother's milk should be changed to mother's own milk

Line 31: vegetative form should be changed to vegetative forms

Line 50: needing not needed

Line 51: necessary procedures, not necessity procedures

Line 57: "to analyze current the best characterized methods" should be changed to: "to analyze the best characterized and current methods"

lLne 75: homeostasis not homeostatis

Line 77: from newborn mouth cavity change to: from the newborn's mouth cavity

Line 123: ultrasound not ultrasounds

Lines 124-126: the verb is missing

Line 142: to before: change to: before...

Line 159: find, not found

Line 171: A recent study, not "Recent study can be an effective method..."

Line 179: different types, not type

Line 202: that were used

Line 262: but latter study: do you mean the latter study or a later study by this same group? It is not clear.

Line 263: the earlier studies' findings are described in the past tense but this study, the findings are described in the present tense. Please make all past study findings in the past tense.

line 269-270: decimal points where commas are used: e.g. 88,23 should be 88.23

unclear what "protective mechanisms of a body" means. Please clarify.

Line 291: statistically significant differences not difference

Table 2: need header appearing on each new page; otherwise, the reader has to scroll up to the beginning to be reminded of what appears in each column

Line 302: from the food industry, not from food industry

Lines 302-303: change to: to minimize losses as a consequence of human milk processing

 Line 306: change to: if there is a real possibility to introduce...

Author Response

We agree with suggestions of the Reviewer 1. All the indicated changes have been applied 

( marked in green in the resubmitted manuscript)

Reviewer 2 Report

Thanks to the authors for this interesting review of techniques.

1) Minor English "Donor milk is a recommended as an alternative". I believe a should be stricken.

2) The last added sentences in materials and methods regarding methodologic discrepancies and study limitations should be in the discussion section.

3) Minor English revision: Microbiota associated to human milk mostly reflects maternal health status but contributing both, mother and infant homeostasis. Correct to" contributes to both maternal and infant homeostasis".

4) The article would benefit from headings such as implications of different pasteurization methods on:

lactoferrin, lactoperoxidase and lysozyme

Oligosaccharides

Fat contents

Vitamins

Growth factors

Etc. This would help focus the reader on specific issues.

5) The conclusions section is very brief. It would be helpful for the authors to provide a bit more synthesis of the research they have detailed. What further research is needed before there should be a shift in technology? Are certain technologies better suited for certain applications, etc.

Author Response

We modified our manuscript according to the reviewer‘ minor suggestions (1-3). Thank you for draw our attention to make paper easier to reading (4). We hope improved it by divided the section 3.3 into the additional paragraphs focusing on specific issue.
The conclusion section was rewarded by adding additional threads in resubmitted manuscript (5)